# Recent Progress in Constructing Plasmonic Metal/Semiconductor Hetero-Nanostructures for Improved Photocatalysis

**Liang Ma [1],\*** , **Shuang Chen [1]**, **Yun Shao [2]**, **You-Long Chen [1]**, **Mo-Xi Liu [1]**, **Hai-Xia Li [1]**, **Yi-Ling Mao [1] and Si-Jing Ding [3],\***

[1] Hubei Key Laboratory of Optical Information and Pattern Recognition, Wuhan Institute of Technology, Wuhan 430205, China; Chenshuang0118@126.com (S.C.); chenyoulong2@126.com (Y.-L.C.); liumoxi00@126.com (M.-X.L.); lihai198505@126.com (H.-X.L.); Maoyiling826@126.com (Y.-L.M.)

[2] School of Material Science and Engineering, Wuhan Institute of Technology, Wuhan 430205, China; shaoyun_wit@163.com

[3] School of Mathematics and Physics, China University of Geosciences (Wuhan), Wuhan 430074, China

\* Correspondence: maliang@wit.edu.cn (L.M.); dingsijing@cug.edu.cn (S.-J.D.)

**Abstract:** Hetero-nanomaterials constructed by plasmonic metals and functional semiconductors show enormous potential in photocatalytic applications, such as in hydrogen production, $CO_2$ reduction, and treatment of pollutants. Their photocatalytic performances can be better regulated through adjusting structure, composition, and components' arrangement. Therefore, the reasonable design and synthesis of metal/semiconductor hetero-nanostructures is of vital significance. In this mini-review, we laconically summarize the recent progress in efficiently establishing metal/semiconductor nanomaterials for improved photocatalysis. The defined photocatalysts mainly include traditional binary hybrids, ternary multi-metals/semiconductor, and metal/multi-semiconductors heterojunctions. The underlying physical mechanism for the enhanced photocatalysis of the established photocatalysts is highlighted. In the end, a brief summary and possible future perspectives for further development in this field are demonstrated.

**Keywords:** photocatalysis; plasmon; metal/semiconductor; electron transfer; energy conversion

## 1. Introduction

The increasing energy shortage and environmental crisis are common problems faced by the world in the twenty-first century, and the whole world is seeking ways to ease these issues. As a clean technique to convert solar energy, semiconductor-based photocatalysis has been intensely investigated and widely used in energy stores and pollutant treatment [1–10]. Since Fujishima and Honda firstly applied $TiO_2$ electrode to electrochemical water splitting under ultraviolet light irradiation in 1972 [11], large amounts of semiconductor materials have been extensively explored for diverse photocatalytic applications, such as pollution degradation, water splitting, $CO_2$ reduction. [12–18]. In the case of the photocatalytic process, semiconductors can only absorb a photon whose energy is equal or larger than its band gap. However, the commonly used semiconductors often have wide band gaps, indicating that they could only be motivated by ultraviolet light [19,20]. As the semiconductor is excited, the formation of photogenerated electron–hole pairs occurs, while the fast recombination of electron–hole pairs in a semiconductor plays a negative role on the photocatalytic reaction. Therefore, to optimize the photocatalytic performance of semiconductor, the exploring of new strategies, to expand the light response region and accelerate the separation rate of electron–hole pairs, is necessary.

Over the past two decades, metal nanocrystals have attracted intense research attention due to their extraordinary physical and chemical characteristics [21–23]. The most fascinating property of metal nanocrystals is their plasmonic optical peculiarity. Plasmon resonance of metal nanomaterials refers to the collective oscillations of light-excited free charges, which can endow metals with strong light absorption and scattering cross-sections [24–30]. Furthermore, the localized surface plasmon resonance excitation of a metal nanocrystal can induce a strong local electromagnetic field. Moreover, upon resonant excitation, plasmonic metals interact strongly with incident light, the oscillation of free electrons dephases in a short time, then generates energetic hot electrons and holes. Only certain metal nanocrystals, such as Au, Al, Ag, and Cu, possess noticeable surface plasmon resonance [31–34]. The plasmon resonance of these metal nanomaterials can be easily adjusted through varying the sizes and morphologies, which could across the entire visible spectrum [35–40]. With these characters, the plasmonic metals deservedly display great promises for the effective solar energy conversion in photocatalytic reaction.

Incorporating plasmonic metal nanocrystals with semiconductor photocatalysts to form metal/semiconductor hybrid nanostructures is a potential way to enlarge the light absorption, charge generation, and separation during the photocatalytic process. Great efforts have been applied to construct various metal/semiconductors hybrids with excellent photocatalytic performance [41–45]. In present article, we give a mini-review about the recent efforts in efficient preparation of metal/semiconductor nanomaterials for improved photocatalytic applications. The defined photocatalysts are mainly centralized in traditional binary hybrids, ternary multi-metals/semiconductor, and metal/multi-semiconductors heterojunctions. The underlying physical mechanism (including plasmon coupling of multi-metals, co-catalytic effect of functional metals, plasmon-mediated Z-scheme, and p–n heterojunctions of multi-semiconductors) for the efficiently improved photocatalysis of metal/semiconductor heterojunctions are highlighted. In the end, a concise summary and discussion on the future challenges in the area of metal/semiconductor photocatalysis are demonstrated.

## 2. Binary Metal/Semiconductor Hybrids for Enhanced Photocatalysis

Binary photocatalysts constructed by one plasmonic metal and one semiconductor are the most frequently studied. In a metal/semiconductor heterojunction, plasmon can modulate photocatalysis mainly through the following characteristics: (1) strong light absorption and scattering; (2) large local electromagnetic field; (3) abundant hot electrons generation [46–50]. Thus, in the photocatalytic process, plasmon could promote the redox reaction via the following pathways: enlarging light trapping, speeding charge separation, hot electron injection, and plasmon-induced energy transfer [51–55]. Since the optical and photocatalytic performances of metal/semiconductor hybrids are highly dependent on their morphologies and structures, the design of hetero-nanostructure is very important. In this section, we mainly focus on the recent progress about the structural adjustment of binary metal/semiconductor photocatalysts. Traditional core–shell, yolk–shell, and anisotropic morphologies are highlighted. The underlying enhanced mechanism for photocatalysis of the classified nanostructures is demonstrated.

A typical core–shell structural motif of metal/semiconductors has special advantages for photocatalytic reactions, such as protecting the metal core and enlarging the active sties [56,57]. Generally, the plasmonic metal cores mainly include Au, Ag, Cu, and semiconductor nanoshells that are usually centered in metallic oxide, sulfide, and selenide [58–63]. For instance, Zhang's group reported the synthesis of high-quality Au@CdS core–shell hybrids with atomically organized interfaces [64]. As shown in Figure 1a,b, the high-yield and monodisperse Au@CdS, with quasi-single crystalline shell, was observed. The photocatalytic performance was evaluated by testing the hydrogen generation rate when $Na_2SO_3$ and $Na_2S$ were used as sacrificial agents. The results indicated that Au35@CdS5 (the Au core size was around 35 nm and CdS shell thickness was 5 nm) exhibited the highest activity, with the rate reaching to 24.0 mmol $g^{-1}$ $h^{-1}$ (see Figure 1c). The results of transient

absorption indicated that the specific interfacial characteristic activated efficient hot electrons, injected from Au to CdS, which could be the reason for the excellent photocatalytic activity.

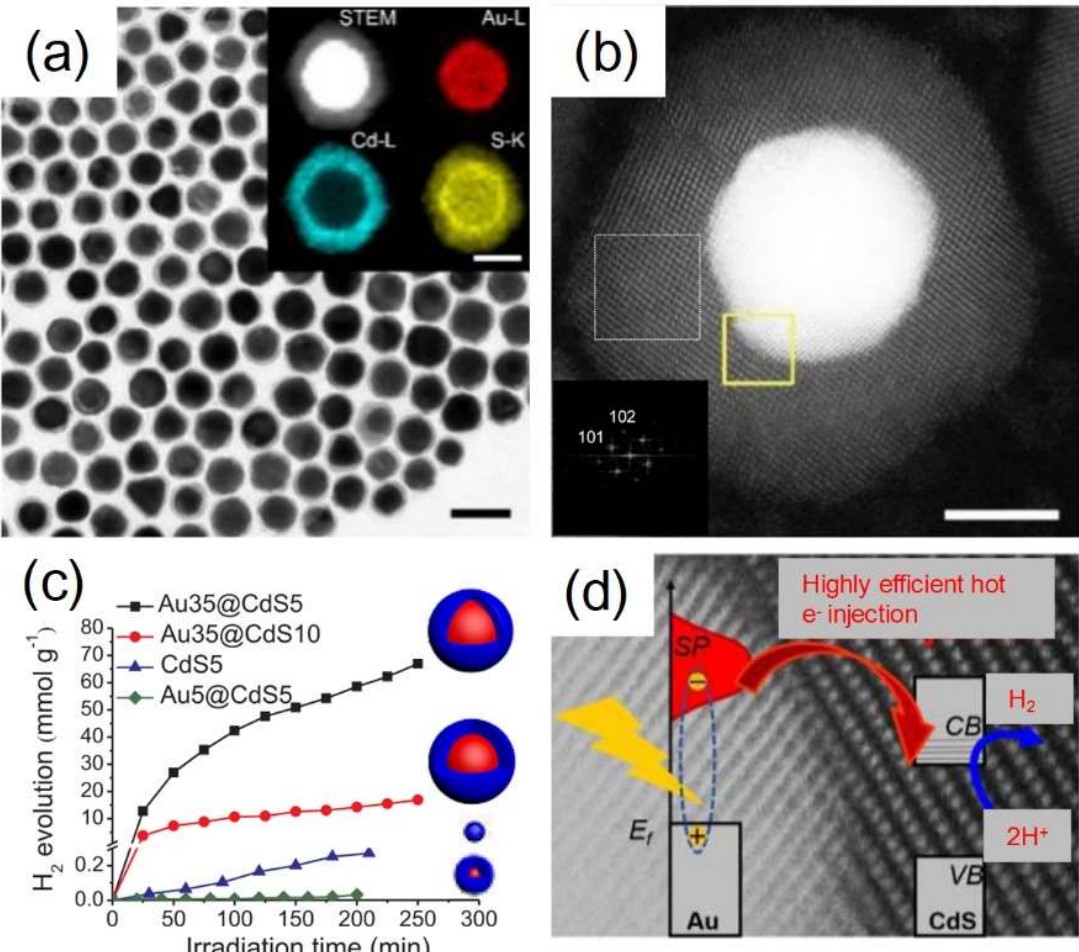

**Figure 1.** (**a**,**b**) Morphology characterization of prepared Au@CdS nanocrystals. Scale bar in (**a**) and (**b**) are 20 nm and 10 nm, respectively. (**c**) Photocatalytic hydrogen evolution of prepared Au@CdS with different architectural feature in comparison with pure CdS under visible light irradiation. (**d**) Possible physical mechanism of the hot electron-mediated photocatalysis. Copyright 2018 Elsevier.

Apart from core–shell nanostructures, metal-semiconductor yolk–shell nanostructures also benefit photocatalysis. The unique void between metal and semiconductor could efficiently improve light trapping and accelerate plasmon-induced resonant energy transfer [65–68]. For example, Han and co-workers reported a yolk–shell nanostructure consisting of plasmonic Au nanorod yolk and CdS shell [69]. The synthetic route was shown in Figure 2a, and the most important point in the process was the action exchange. The identified yolk–shell structure of Au/CdS was given in Figure 2b. The photocatalytic activity was evaluated by testing the hydrogen generation rate when $Na_2SO_3$ and $Na_2S$ were served as sacrificial agents under visible light ($\lambda > 400$ nm) irradiation. The prepared Au–CdS yolk–shell hybrids exhibited largely enhanced photocatalytic activity, as compared to other contrast nanostructures (see Figure 2c). The cooperative effect between multiple photon reflections and the radiative relaxation of plasmon energy were thought to be the promotional effect for the photocatalysis, as proposed in Figure 2d.

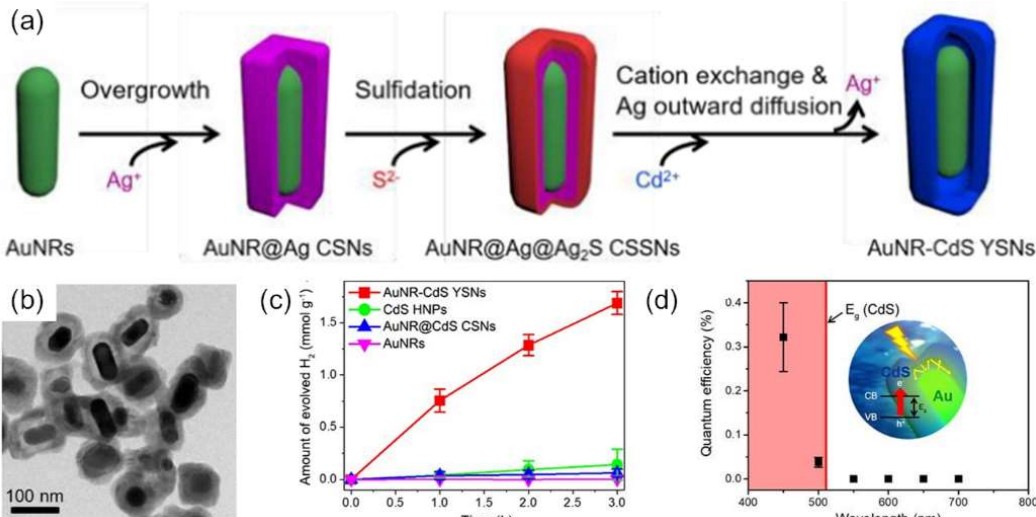

**Figure 2.** (**a**) Schematic illustrating the synthesis route of Au–CdS yolk–shell nanostructures. (**b**) Transmission electron microscopy (TEM) image of Au–CdS yolk–shell nanostructures. (**c**) Rate of evolved hydrogen with different catalysts. (**d**) Hydrogen evolution activity of Au–CdS yolk–shell hybrids as a function of excitation wavelength. Inset shows the schematic illustration of the synergism. Copyright 2018 Royal Society of Chemistry.

In addition, anisotropic binary metal/semiconductor hybrids have unique merits for photocatalytic reaction. By selectively growing semiconductor shells on the surface of metal nanocrystals, the processes of electron injection and energy transfer can be prominently accelerated [70–72]. In 2016, Stucky's group prepared an anisotropic $Au/TiO_2$ nanodumbbell, where the $TiO_2$ nanoshells were spatially grown at the two ends of Au nanorod (see Figure 3a) [73]. Through testing the hydrogen production rate by evaluating the photocatalytic oxidation of methanol, the designed $Au/TiO_2$ showed the best photocatalytic activity (see Figure 3b). As proposed in Figure 3c, for the nanodumbbells, the oxidation occurs on their side surface, the lateral side of Au can be directly used as the electron donor. With Au partially exposed, Au could generate a large local electric field that focused energy transfer to $TiO_2$, thereby enhancing photocatalytic activity. Similarly, Han and co-workers reported the site-selective growth of $Cu_2O$ nanoshells on desired sites of hexoctahedral Au nanocrystals [74]. The morphology characterization in Figure 3d clearly shown that $Cu_2O$ grew on the vertices of hexoctahedral Au ($Au_{vertex}$–$Cu_2O$). $Au_{vertex}$–$Cu_2O$ exhibited excellent photocatalytic activity for hydrogen generation, compared to the other nanostructures. A series of experiments (including adding insulating layer and theoretical simulation) indicated that the strong plasmon resonance, subsequent sustainable hot electron and energy transfer process (1 and 2, labeled in Figure 3f) were responsible for the enhanced photocatalytic activity.

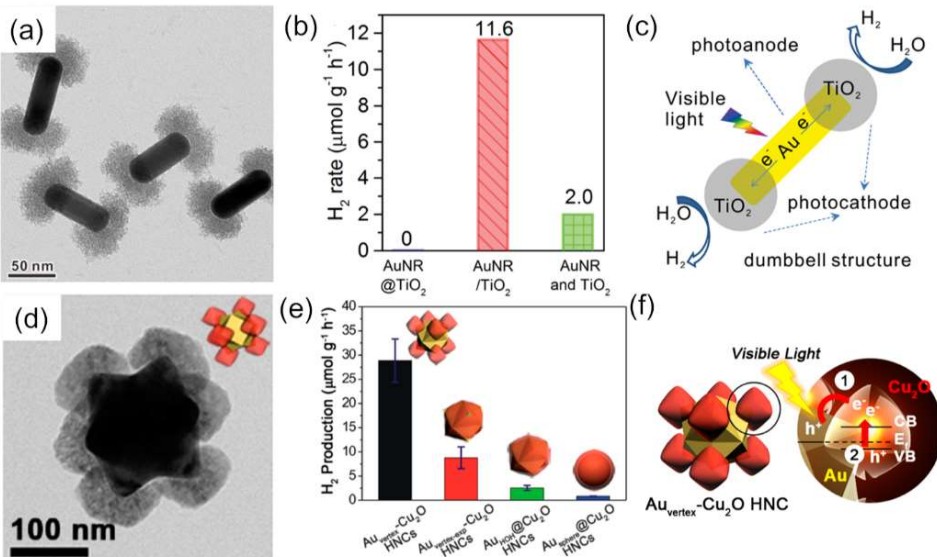

**Figure 3.** (**a**) TEM image of Au/TiO$_2$ nanodumbbells. (**b**) Hydrogen evolution rate by various catalysts. (**c**) Possible photocatalytic mechanism of an Au/TiO$_2$ dumbbell under visible light irradiation. Copyright 2016 American Chemical Society. (**d**) TEM image of Au$_{vertex}$–Cu$_2$O heteronanocrystal. (**e**) Photocatalytic hydrogen generation rates of different heteronanocrystals. (**f**) Schematic illustration the possible charge transfer in Au$_{vertex}$–Cu$_2$O hybrids. Copyright 2016 American Chemical Society.

## 3. Plasmon Coupling, Co-Catalytic Effect, and Components Arrangement in Ternary Multi-Metals/Semiconductor Catalysts

Integrating one metal nanocrystal with binary metal/semiconductor to form ternary hetero-nanostructure could dramatically improve the photocatalytic activity, due to the synergistic effect between the three nano-spaced components [75,76]. The introduced metals could be a plasmonic donor or functional catalyst. In that way, the synergistic effect includes plasmon coupling and co-catalytic effect. Furthermore, the arrangements of the three different components also influence the whole photocatalytic performance. In the current section, we focus on the progress achieving in constructing ternary multi-metals/semiconductor photocatalysts. The excellent photocatalytic performances of defined nanomaterials and underlying physical mechanism are highlighted.

The plasmon coupling between two metals could generate excellent optical properties. On the one hand, the light trapping of multi-metals can be largely enhanced due to the strong plasmon coupling of two types of metals [77,78]. On the other hand, the local electric field of multi-metals can be largely enhanced, which can benefit the hot electron generation and electron–hole pairs' separation [79,80]. Recently, Huang and co-workers used Au–Ag bimetallic nanoparticles to modify ZnO nanorods via a simple photodeposition procedure (see Figure 4a) [81]. Through testing the rate of ethylene-oxidation, the ZnO co-decorated by 0.8 wt% Au–Ag exhibited the best photocatalytic activity. The efficient light trapping and fast carrier separation, which were caused by cooperative effect of plasmonic Au–Ag alloys, were thought to be the enhanced factor. Besides, Kamimura et al. prepared (core@shell)@shell ((Au@Ag)@Au) nanocrystals as photosensitizers of TiO$_2$ (see Figure 4c) [82]. They confirmed that the as-prepared catalysts could oxidize 2-propanol, and its activity was about 15× higher when compared with Au/TiO$_2$. The strong local surface plasmon resonance of (Au@Ag)@Au and efficient electrons injection to TiO$_2$ were thought to be the promotional effect.

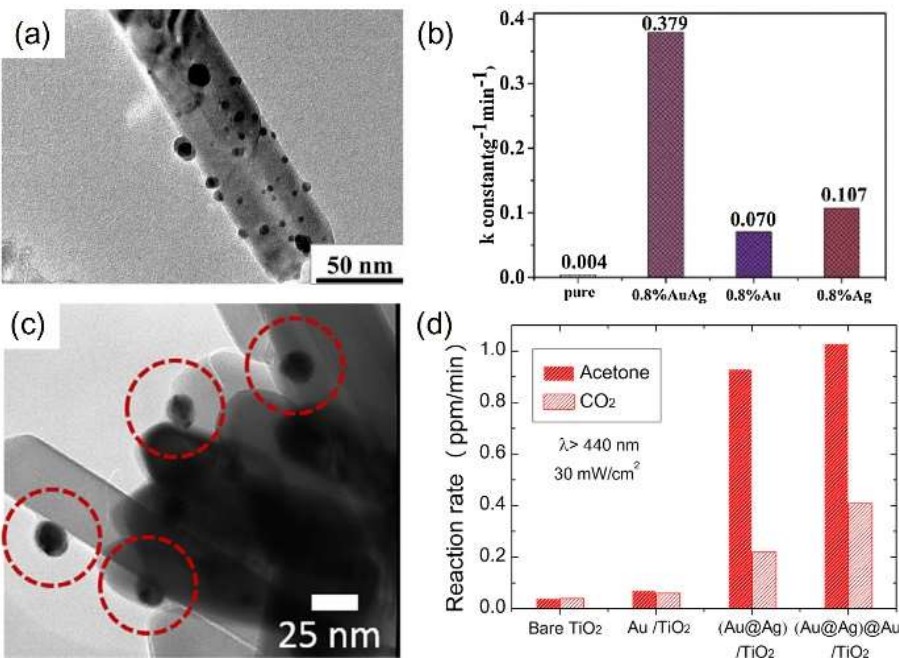

**Figure 4.** (**a**) TEM image of AuAg/ZnO. (**b**) Photocatalytic rate of different catalysts. Copyright 2017 Elsevier. (**c**) TEM image of (Au@Ag)@Au decorated TiO$_2$. (**d**) Photocatalytic performance of the contrast samples under visible light irradiation. Copyright 2017 Elsevier.

As is well-known, Pt and Pd nanocrystals are ideal catalysts for oxygen reduction reactions [83,84]. Commonly, they can be directly decorated on semiconductors, which could be used as co-catalysts and active sites for the improved photocatalysis. Combining Pt or Pd with plasmonic metal/semiconductor hetero-nanostructures can lead to an extreme boost of electron migration in the redox reaction [85–90]. In 2015, Moskovits' group fabricated a device including Au nanorod, TiO$_2$, and Pt, to achieve panchromatic photoproduction of hydrogen [91]. Firstly, Au nanorods with different aspect ratios were dropped cast on quartz slides with panchromatic absorption. Then, TiO$_2$ film was deposited, and played the role of a hot electron filter, and Pt nanoparticles were capped, functioning as the hydrogen evolution catalyst (see Figure 5a). The photocatalytic results revealed that the catalyst with Au nanorods of aspect ratio 1.4 and 3.0 showed the highest photocatalytic activity for hydrogen production.

The arrangement of components in multi-metals/semiconductor heterojunctions has a great influence on their final photocatalytic performance. Properly optimizing the structural configuration could increase the utilization efficiency of plasmon-induced hot electrons [92–94]. Meanwhile, abundant pathways for electron transfer could expedite the separation of electron–hole pairs. Our previous work had certified the important role of structural arrangement for improving the photocatalytic activity [95]. We prepared an each-contacted Au–Pt–CdS hybrid, which is shown in Figure 6a. The photocatalytic testing indicated that Au–Pt–CdS exhibited excellent photocatalytic performance for hydrogen production. Through testing the ultrafast time-resolved transient absorption (see Figure 6c), the multipathway electron-transfer processed in Au–Pt–CdS hybrids could be attested, which were illustrated in Figure 6d. The multiple effective electron-transfer pathways (Au to CdS, Au to Pt, and CdS to Pt) were thought to be the enhancing factor. Similar nanostructures were reported by Dong and co-workers [96]. They synthesized an Au–Pt–CdS photocatalyst, which featured an Au nanorod with tip-coated Pt and side-coated CdS. The as-prepared Au–Pt–CdS possessed efficient UV–vis–NIR-driven plasmonic photoactivity for hydrogen generation. The rational arrangement of the components in Au–Pt–CdS, induced to resonance energy and hot electron transfer, was thought be responsible for the outstanding photocatalytic activity.

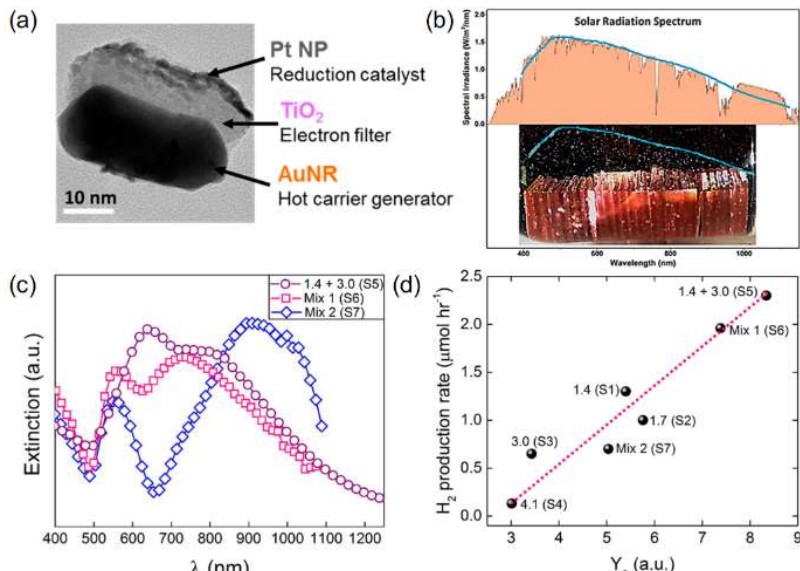

**Figure 5.** (**a**) Typical morphology of an Au–TiO$_2$–Pt nanostructure. (**b**) A digital photograph of hydrogen production. (**c**) Extinction spectra of Au nanorods mixing with different aspect ratios. (**d**) The hydrogen production rates of different catalysts. Copyright 2015 American Chemical Society.

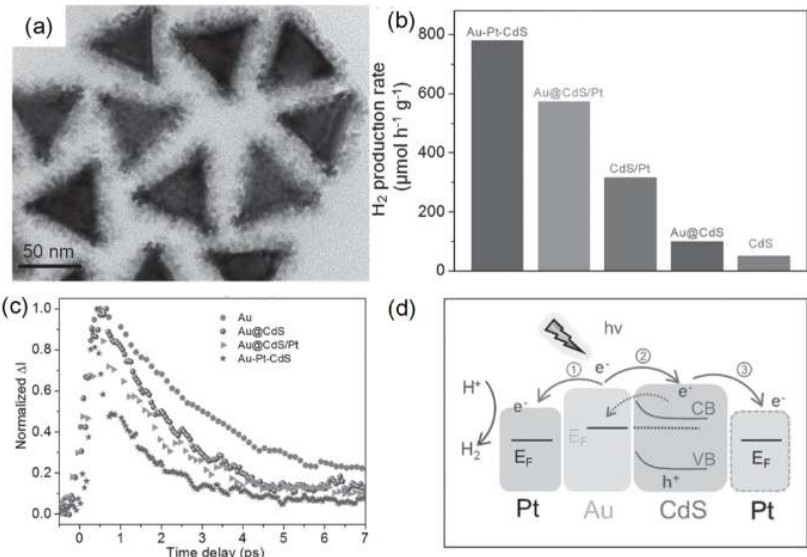

**Figure 6.** (**a**) TEM image of Au–Pt–CdS hybrids. (**b**) Photocatalytic hydrogen production of contrast samples. (**c**) Normalized time-resolved optical differential transmission of different samples. (**d**) Schematic illustration of the possible electron transfer process of Au–Pt–CdS. Copyright 2016 Wiley-VCH.

## 4. Plasmon-Mediated Heterojunctions Photocatalysis in Ternary Metal/Multi-Semiconductors Hetero-Nanostructures

Properly engineered semiconductor heterojunction photocatalysts are proved to show better photocatalytic performance owing to the spatial separation of photo-generated electron–hole pairs. The direct Z-scheme and p–n heterojunctions are the most frequently studied due to their distinct advantages for charge migration [97–100]. By introduction of plasmonic metal with these junctions, the separation rate of electron–hole pairs can be observably expedited. In present section, we briefly review the recent works centered on plasmon-modified Z-scheme and p–n heterojunctions for efficient photocatalysis.

The Z-scheme concept was raised by Bard and co-workers, in order to enlarge the redox potential of the semiconductor heterojunction [101]. Several advantages have been obtained for the Z-scheme configuration, such as effective charge separation, high reduction, and oxidation power, and more participant photocatalysts [102–104]. By introduction of plasmonic metals into this system, the high concentration of hot electrons and large utilization of light induced by plasmon resonance could further promote the photocatalytic reaction [105–107]. For instance, Tang and co-workers reported a plasmon-excited dual Z-scheme system in ternary $BiVO_4/Ag/Cu_2O$ nanocomposite [108]. Specifically, the $Cu_2O$ particles were coupled with $BiVO_4$ nanostructure, which were deposited with metallic Ag nanocrystals. The specific morphology of $BiVO_4/Ag/Cu_2O$ was shown in Figure 7a. It was shown that the ternary $BiVO_4/Ag/Cu_2O$ had the highest photocatalytic activity (see Figure 7b). The enhanced photocatalytic activity was mainly caused by the collaboration of $Cu_2O$, Ag, and $BiVO_4$, especially the plasmon-mediated dual Z-scheme, and strong local electric field induced by Ag, which was proposed in Figure 7c.

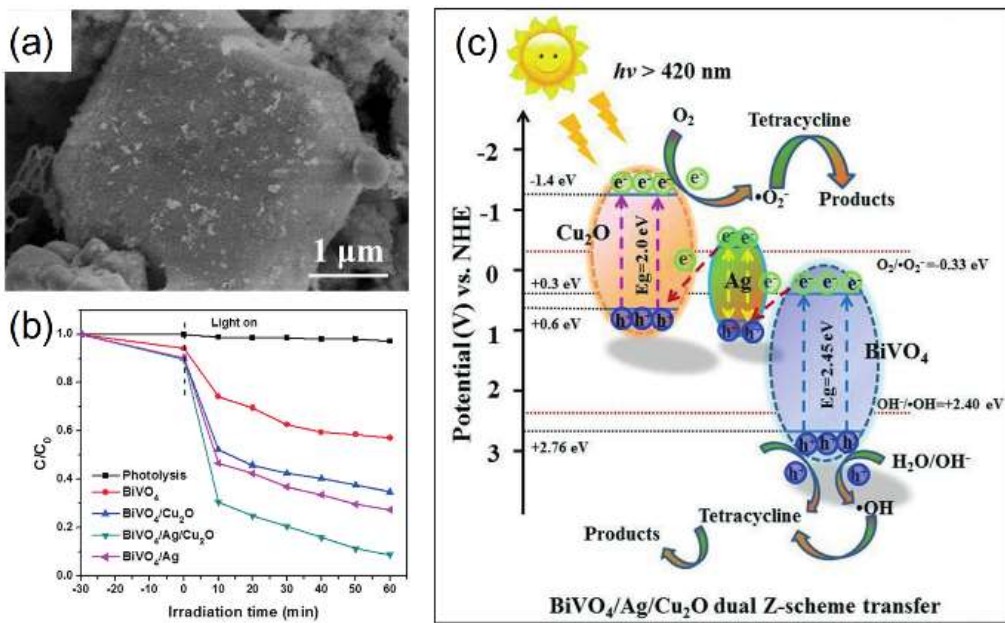

**Figure 7.** (**a**) Morphology characterization of $BiVO_4/Ag/Cu_2O$ composites. (**b**) The photocatalytic performance of different photocatalysts. (**c**) Possible reaction mechanism in the $BiVO_4/Ag/Cu_2O$ nanocomposite-based reaction systems. Copyright 2017 Royal Society of Chemistry.

The photocatalytic concept of p–n junction was put forward to speed up the electron–hole transfer along the junction through introducing an additional electric field [109–111]. Generally, to obtain an effective p–n heterojunction photocatalyst, we can simply combine a p-type semiconductor with a n-type semiconductor. As the p–n heterojunction photocatalyst is excited by incident photons, both types of semiconductor can generate electron–hole pairs. Then, the electrons and holes will be transferred under the effect of the internal electric field. Typically, the electrons will migrate to the conduction band of the n-type semiconductor, and the holes migrate to the valence band of the p-type semiconductor, thus resulting in the fast spatial separation of the electron–hole pairs [112,113]. Combing plasmonic metal nanocrystals with p–n heterojunction could further speed the process of electron–hole pairs' separation [114,115]. For instance, Zhou et al. fabricated a unique three-dimensional $Cu_xO/ZnO@Au$ composite, in which CuO nanowires were coupled with ZnO nanodisks and Au nanoparticles (see Figure 8a–c) [116]. The photocatalytic results indicated that $Cu_xO/ZnO@Au$ exhibited the highest hydrogen production rate, which reaches to 12.4 μmol cm$^{-2}$ h$^{-1}$. The possible enhanced mechanism was proposed in Figure 8e. On the one hand, the cooperative effect between CuO, ZnO, and the formed p–n junctions could enlarge the light utilization. Furthermore, the

plasmon-mediated p–n junction could accelerate the separation of electron–hole pairs, thus enhancing the photocatalytic activity.

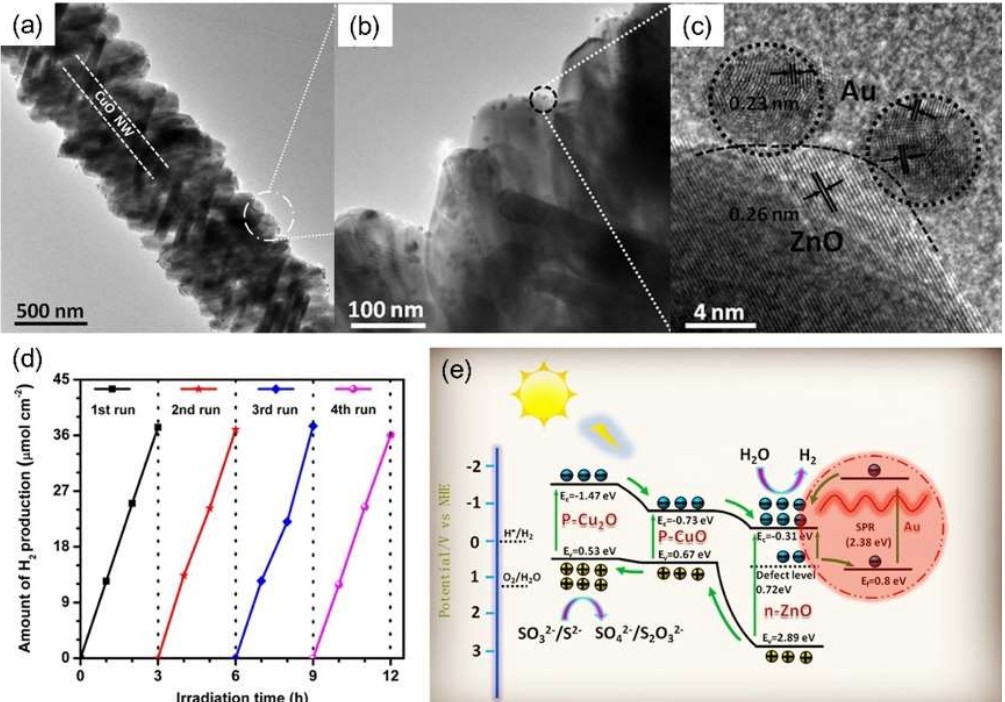

**Figure 8.** (**a**–**c**) Morphology and compositions characterizations of $Cu_xO/ZnO@Au$ hetero-nanostructures. (**d**) Recycling photocatalytic test of $Cu_xO/ZnO@Au$. (**e**) Schematic illustration of photoexcited carrier dynamics in $Cu_xO/ZnO@Au$. Copyright 2015 American Chemical Society.

## 5. Conclusions and Outlook

In recent years, the research focused on plasmon-mediated photocatalysis of hetero-nanomaterials has been steadily expanding, and the progress mentioned in the present work is only a drop in the bucket. In this article, we present a concise review of the recent progress in the field of plasmon-mediated metal/semiconductor photocatalysts, including their designs, synthesis, and photocatalytic applications. The physical mechanism for the enhanced photocatalysis, in binary and ternary metal/semiconductor heterojunctions, is briefly classified.

Although many exciting results have been achieved in this field, the practical efficiency of the photocatalytic reaction is still low, and the further industrialization and commercialization of photocatalysts requires untiring exploration. In our option, the future research effort in this field may be centered on the following aspects. Firstly, the synthetic method is the footstone for the preparation of photocatalysts. Currently, efforts should be continued in exploring facile, economic, environmentally friendly strategies for preparing high-performance metal/semiconductor photocatalysts. Secondly, searching new plasmonic materials and functional semiconductors to form effective heterojunctions is one of the key research goals. More research efforts should be centered on the developing new materials with low cost and high solar-conversion efficiency. Thirdly, the mechanism of plasmon-enhanced photocatalytic activity requires further systematic studies. An effective ultrafast spectral analysis technique should be applied to quantitatively verify the migration pathway of charge. Meanwhile, the corresponding theoretical calculations and modeling methods for optimizing the structural construct and charge transfer should receive more attention. Further theoretical achievement could offer a better understanding of the charge and energy transfer kinetics, thereby guiding the design of high-quality photocatalyst.

In brief, we have reviewed the recent progress in constructing metal/semiconductor photocatalysts for improved photocatalytic applications. The variation of their photocatalytic performance with structural adjustment and component arrangement was demonstrated. We hope this article will inspire the further design and fabrication of functional materials which could be used in photocatalysis and other important applications.

**Funding:** This research was funded by the National Natural Science Foundation of China (11804257), Natural Science Foundation of Hubei Province (2018CFB106), college president's foundation (2018130), and scientific research foundation (18QD24 and 18QD25) of Wuhan Institute of Technology.

**Conflicts of Interest:** There are no conflicts to declare.

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
