# Peer review of "Recent Progress in Constructing Plasmonic Metal/Semiconductor Hetero-Nanostructures for Improved Photocatalysis"

_catalysts, doi:10.3390/catal8120634_

Round 1
Reviewer 1 Report
I think it is a well written review. However, I think that the range of reviews is too limited if there are only two types of metal mentioned (Au and Pt).
I would also like to comment on nanostructure made of other metals such as Pd, Ag and Cu.
Author Response
Point 1: I think it is a well written review. However, I think that the range of reviews is too limited if there are only two types of metal mentioned (Au and Pt).
I would also like to comment on nanostructure made of other metals such as Pd, Ag and Cu.
Response 1: Thank the reviewer for the positive evaluation. As you said, some other noble metal nanocrystals, such as Ag, Cu and Pd, have also been widely used in photocatalysis. We have added some recent works in the references and appended relative discussion in the main text.
Revision in main text (Line 83, Page 2)
“Generally, the plasmonic metal cores mainly include Au, Ag, Cu, and semiconductor nanoshells usually are centered on metallic oxide, sulfide, and selenide [58-63].”
Revision in main text (Line 167, Page 5)
“As is well-known, Pt and Pd nanocrystals are ideal catalysts for oxygen reduction reaction [83-84]. Commonly, they can be directly decorated on semiconductor which could be used as co-catalysts and active sites for the improved photocatalysis. Combine Pt or Pd with plasmonic metal/semiconductor hetero-nanostructures can extremely boost the redox reaction [85-90].”
Added references:
“62. Li, J., Cushing, S. K., Bright, J., Meng, F., Senty, T. R., Zheng, P. Bristow, A. D., Wu, N. Ag@ Cu2O core-shell nanoparticles as visible-light plasmonic photocatalysts. ACS Catal. 2012, 3, 47-51.
63. Ai, Z., Zhang, L., Lee, S., Ho, W. Interfacial hydrothermal synthesis of Cu@Cu2O core-shell microspheres with enhanced visible-light-driven photocatalytic activity. J. Phys. Chem. C 2009, 113, 20896-20902.
89. Su, R., Tiruvalam, R., Logsdail, A. J., He, Q., Downing, C. A., Jensen, M. T., Dimitratos, N., Kesavan, L., Wells, P. P., Bechstein, R., Jensen, H. H., Wendt, S., Catlow, C. R. A., Kiely, C. J., Hutchings, G. J., Besenbacher, F. Designer titania-supported Au–Pd nanoparticles for efficient photocatalytic hydrogen production. ACS Nano 2014, 8, 3490-3497.
90. Darabdhara, G., Boruah, P. K., Borthakur, P., Hussain, N., Das, M. R., Ahamad, T., Alshehri, S. M. Malgras, V., Wu, K. C. W., Yamauchi, Y. Reduced graphene oxide nanosheets decorated with Au–Pd bimetallic alloy nanoparticles towards efficient photocatalytic degradation of phenolic compounds in water. Nanoscale 2016, 8, 8276-8287.”

Reviewer 2 Report
In this review, the authors report the most recent progress on plasmonic metal-semiconductor nanostructures for photocatalytic applications. The topic is well-covered, and the authors provide adequate revision of some of the most appealing structures, although probably some of the mechanistic aspects could be explained more in detail. In fact, some of the possible structures, combining the different roles of plasmonic metals, catalysts and semiconductors are presented, but only in the last section, the authors mention the challenges of plasmon-mediated photocatalytic processes. So, besides the presentation of such appealing nanostructures, an additional section addressing such challenges and a comparison of the different alternative nanostructures would improve the overall merit of the review.
Also, I would suggest to provide better details on experimental conditions. For instance, in some of the reported results, it is not clear which is the oxidation reaction.
Finally, I have some specific comments:
-In Figure 5c, how is the extinction coefficient "connected" to the photocatalytic activity?
-Please correct line 234: it is Figure 7, not 9.
-In Figure 7, what do S0, S1, S2... represent?
-Please correct in line 269-270, "conduction" and "valence" band. Also, the paragraph in lines 263-272 can be put before, as the BiVO4/Cu2O heterostructure described before is also a n-p heterojunction.
-In Figure 9, which samples are CZ1, CZ2...?
-English must be more carefully checked.
Author Response
Point 1: In this review, the authors report the most recent progress on plasmonic metal-semiconductor nanostructures for photocatalytic applications. The topic is well-covered, and the authors provide adequate revision of some of the most appealing structures, although probably some of the mechanistic aspects could be explained more in detail. In fact, some of the possible structures, combining the different roles of plasmonic metals, catalysts and semiconductors are presented, but only in the last section, the authors mention the challenges of plasmon-mediated photocatalytic processes. So, besides the presentation of such appealing nanostructures, an additional section addressing such challenges and a comparison of the different alternative nanostructures would improve the overall merit of the review.
Response 1: Thanks very much for your valuable comments. In this mini-review, we mainly focus on the strategies of constructing metal/semiconductor photocatalysts. The underlying physical mechanism for the improved photocatalysis is briefly classified. The nanostructure mentioned in this work is only a small part, we just list some representative parts. To date, the field of plasmon-enhanced photocatalysis is still limit to fundamental research. The challenges for the further industrialization and commercialization of the photocatalysts is comprehensive. Developing new synthetic method, exploring the new functional materials as well as the investigation of physical mechanism is necessary to promote the advance of this field. As you said, the comparison of the photocatalytic performances of the mentioned photocatalysts is helpful for the further understanding the progress achieved in this field. However, the photocatalytic reactions of these photocatalysts are diverse, such as hydrogen production, CO2 reduction, pollutants treatment. It’s too hard to definite the same standards to compare their photocatalytic activity because of it involves a lot of reaction conditions.
Point 2: Also, I would suggest to provide better details on experimental conditions. For instance, in some of the reported results, it is not clear which is the oxidation reaction.
Response 2: We have added some relative experimental conditions of the reported results in the main text.
Point 3: Finally, I have some specific comments:
-In Figure 5c, how is the extinction coefficient "connected" to the photocatalytic activity?
Response 3: Figure 5c shown the extinction spectra of Au nanorods with different aspect ratios. We have corrected the figure caption in the main text.
Point 4: Please correct line 234: it is Figure 7, not 9.
-In Figure 7, what do S0, S1, S2... represent?
Response 4: We have corrected and added the expression in the main text.
Point 5: Please correct in line 269-270, "conduction" and "valence" band. Also, the paragraph in lines 263-272 can be put before, as the BiVO4/Cu2O heterostructure described before is also a n-p heterojunction.
In Figure 9, which samples are CZ1, CZ2...?
English must be more carefully checked.
Response 5: Thanks very much for your kinder reminder, we have checked the manuscript carefully and modified the incorrect expression in the main text. We have revised the figure to make it looks more clearly.

Reviewer 3 Report
This mini-review describes some recent advancements on metal/semiconductor heterojunction photocatalysts, including designs, synthesis, applications and photocatalytic mechanisms.
In my opinion, notwithstanding the review is partial and not comprehensive, it is quite interesting and stimulates further research work in this important field. Therefore I think that it should be accepted for publication in Catalysts after the following minor revisions have been made:
- Along the paper several misprints are present. Hereinafter some examples: lines 20 and 64: is instead of are; pag.2, line 48: dephases and leads instead of dephase and lead; line 74 dependent instead of depended; line 84: …. indicated that Au35@CdS5 which the Au core size was maintained ….. is difficult to understand, lines 153-155 are also incorrect; line 229: by introducing instead of by introduce; and so on. Therefore I suggest a more careful control of the text and the correction of all misprints.
- Line 13: I think it is better to remove water splitting, in so as it is included in the more general hydrogen production
- Line 14: Probably composition should be a more proper word instead of ingredient
- Line 120: are the authors sure that hydrogen was produced by degradation (oxidation ?) of methanol ? Please specify or correct.
- Finally, from a graphical point of view I must note that some figures (for instance 4b, 7j, 8d) appear too small to be readable and therefore must be somehow rearranged.
Author Response
Point 1: This mini-review describes some recent advancements on metal/semiconductor heterojunction photocatalysts, including designs, synthesis, applications and photocatalytic mechanisms.
In my opinion, notwithstanding the review is partial and not comprehensive, it is quite interesting and stimulates further research work in this important field. Therefore, I think that it should be accepted for publication in Catalysts after the following minor revisions have been made:
Along the paper several misprints are present. Hereinafter some examples: lines 20 and 64: is instead of are; pag.2, line 48: dephases and leads instead of dephase and lead; line 74 dependent instead of depended; line 84: …. indicated that Au35@CdS5 which the Au core size was maintained ….. is difficult to understand, lines 153-155 are also incorrect; line 229: by introducing instead of by introduce; and so on. Therefore I suggest a more careful control of the text and the correction of all misprints.
Response 1: Thank you very much for your valuable comments. We have carefully checked the manuscript and revised the incorrect expression.
Point 2: Line 13: I think it is better to remove water splitting, in so as it is included in the more general hydrogen production
Response 2: We have deleted the “water splitting” in the abstract.
Point 3: Line 14: Probably composition should be a more proper word instead of ingredient
Response 3: We have replaced the “ingredient” by “composition”.
Point 4: Line 120: are the authors sure that hydrogen was produced by degradation (oxidation ?) of methanol ? Please specify or correct.
Response 4: The hydrogen production was produced by oxidation of methanol. We have corrected the expression in the main text.
Point 5: Finally, from a graphical point of view I must note that some figures (for instance 4b, 7j, 8d) appear too small to be readable and therefore must be somehow rearranged.
Response 5: Thanks very much for your kind reminder, we have adjusted the figures to make them look more clearly.
